# Timewarp: Transferable Acceleration of Molecular Dynamics by Learning Time-Coarsened Dynamics

**Leon Klein**[*][†]
Freie Universität Berlin
leon.klein@fu-berlin.de

**Andrew Y. K. Foong**[*]
Microsoft Research AI4Science
andrewfoong@microsoft.com

**Tor Erlend Fjelde**[*][†]
University of Cambridge
tef30@cam.ac.uk

**Bruno Mlodozeniec**[*][†]
University of Cambridge
bkm28@cam.ac.uk

**Marc Brockschmidt**[†]

**Sebastian Nowozin**[†]

**Frank Noé**
Microsoft Research AI4Science
Freie Universität Berlin
Rice University
franknoe@microsoft.com

**Ryota Tomioka**
Microsoft Research AI4Science
ryoto@microsoft.com

## Abstract

*Molecular dynamics* (MD) simulation is a widely used technique to simulate molecular systems, most commonly at the all-atom resolution where equations of motion are integrated with timesteps on the order of femtoseconds ($1\text{fs} = 10^{-15}\text{s}$). MD is often used to compute equilibrium properties, which requires sampling from an equilibrium distribution such as the Boltzmann distribution. However, many important processes, such as binding and folding, occur over timescales of milliseconds or beyond, and cannot be efficiently sampled with conventional MD. Furthermore, new MD simulations need to be performed for each molecular system studied. We present *Timewarp*, an enhanced sampling method which uses a normalising flow as a proposal distribution in a Markov chain Monte Carlo method targeting the Boltzmann distribution. The flow is trained offline on MD trajectories and learns to make large steps in time, simulating the molecular dynamics of $10^5 - 10^6$ fs. Crucially, Timewarp is *transferable* between molecular systems: once trained, we show that it generalises to unseen small peptides (2-4 amino acids) at all-atom resolution, exploring their metastable states and providing wall-clock acceleration of sampling compared to standard MD. Our method constitutes an important step towards general, transferable algorithms for accelerating MD.

## 1 Introduction

Molecular dynamics (MD) is a well-established technique for simulating physical systems at the atomic level. When performed accurately, it provides unrivalled insight into the detailed mechanics of molecular motion, without the need for wet lab experiments. MD simulations have been used to understand processes of central interest in molecular biophysics, such as protein folding [29, 22], protein-ligand binding [1], and protein-protein association [31]. Many crucial applications of MD boil down to efficiently sampling from the *Boltzmann distribution*, *i.e.*, the equilibrium distribution

---

[*]Equal contribution.
[†]Work done while at Microsoft Research.

37th Conference on Neural Information Processing Systems (NeurIPS 2023).

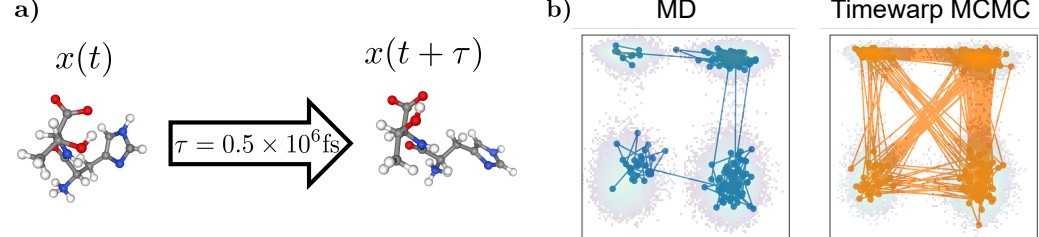

Figure 1: (a) Initial state $x(t)$ (*Left*) and accepted proposal state $x(t+\tau) \sim p_\theta(x(t+\tau)|x(t))$ (*Right*) sampled with Timewarp for the dipeptide HT (unseen during training). (b) TICA projections of simulation trajectories, showing transitions between metastable states, for a short MD simulation (*Left*) and Timewarp MCMC (*Right*), both run for 30 minutes of wall-clock time. Timewarp MCMC achieves a speed-up factor of $\approx 33$ over MD in terms of effective sample size per second.

of a molecular system at a temperature $T$. Let $(x^p(t), x^v(t)) := x(t) \in \mathbb{R}^{6N}$ be the state of the molecule at time $t$, consisting of the positions $x^p(t) \in \mathbb{R}^{3N}$ and velocities $x^v(t) \in \mathbb{R}^{3N}$ of the $N$ atoms in Cartesian coordinates. The Boltzmann distribution is given by:

$$\mu(x^p, x^v) \propto \exp\left(-\frac{1}{k_B T}(U(x^p) + K(x^v))\right), \quad \mu(x^p) = \int \mu(x^p, x^v)\, \mathrm{d}x^v. \tag{1}$$

where $U(x^p)$ is the potential energy, $K(x^v)$ is the kinetic energy, and $k_B$ is Boltzmann's constant. Many important quantities, such as the free energies of protein folding and protein-ligand binding, can be computed as expectations under $\mu(x^p)$. A popular MD method to sample from $\mu(x^p)$ is *Langevin dynamics* [19], which obeys the following stochastic differential equation (SDE):

$$m_i \mathrm{d}x_i^v = -\nabla_i U \mathrm{d}t - \gamma m_i x_i^v \mathrm{d}t + \sqrt{2m_i \gamma k_B T}\mathrm{d}B_t. \tag{2}$$

Here $i$ indexes the atoms, $m_i$ is the mass of atom $i$, $U(x^p)$ is the potential energy, $\gamma$ is a friction parameter, and $\mathrm{d}B_t$ is a standard Brownian motion process. Starting from an initial state $x(0)$, simulating Equation (2), along with the relationship $\mathrm{d}x^p = x^v \mathrm{d}t$, yields values of $x(t)$ that are distributed according to the Boltzmann distribution as $t \to \infty$. Standard MD libraries discretise this SDE with a timestep $\Delta t$, which must be chosen to be $\sim 1\text{fs} = 10^{-15}$s for stability. Unfortunately, many biomolecules contain metastable states separated by energy barriers that can take milliseconds of MD simulation time ($\sim 10^{12}$ sequential integration steps) to cross, rendering this approach infeasible. To overcome this, prior work has produced an array of *enhanced sampling methods*, such as coarse graining [3, 14] and metadynamics [18]. However, these methods require domain knowledge specific to each molecular system to implement effectively.

Standard MD simulations do not transfer information between molecular systems: for each system studied, a new simulation must be performed. This is a wasted opportunity: many molecular systems exhibit closely related dynamics, and simulating one system should yield information relevant to similar systems. In particular, proteins, being comprised of sequences of 20 kinds of amino acids, are prime candidates to study this kind of transferability. We propose *Timewarp*, a general, transferable enhanced sampling method which uses a *normalising flow* [35] as a proposal for a Markov chain Monte Carlo (MCMC) method targeting the Boltzmann distribution. Our main contributions are:

1. We present the first ML algorithm working in general Cartesian coordinates that demonstrates *transferability* to small peptide systems unseen during training.
2. We demonstrate, for the first time, wall-clock acceleration of asymptotically unbiased MCMC sampling of the Boltzmann distribution of unseen peptide systems.
3. We define an MCMC algorithm targeting the Boltzmann distribution using a conditional normalising flow as a proposal distribution, with a Metropolis-Hastings (MH) correction step to ensure detailed balance (Section 3.4).
4. We design a permutation equivariant, transformer-based normalising flow.
5. We produce a novel training dataset of MD trajectories of thousands of small peptides.
6. We show that even when deployed *without* the MH correction (Section 3.5), Timewarp can be used to explore metastable states of new peptides much faster than MD.

The code is available here: `https://github.com/microsoft/timewarp`. The datasets are available upon request[3].

## 2 Related work

There has recently been a surge of interest in deep learning on molecules. *Boltzmann generators* [28, 17, 15] use flows to sample from the Boltzmann distribution asymptotically unbiased. There are two ways to generate samples: (i) Produce i.i.d. samples from the flow and use statistical reweighting to debias expectation values. (ii) Use the Boltzmann generator in an MCMC framework [4], as in Timewarp. Currently, Boltzmann generators lack the ability to generalize across multiple molecules, in contrast to Timewarp. The only exception is [11], who propose a diffusion model in torsion space and use the underlying ODE as a transferable Boltzmann generator. However, in contrast to Timewarp, they use internal coordinates and do not operate in the all atom system. Moreover, [13, 25] recently introduced a Boltzmann generators in Cartesian coordinates for molecules, potentially enabling transferable training. Recently, [46] proposed *GeoDiff*, a diffusion model that predicts molecular conformations from a molecular graph. Like Timewarp, GeoDiff works in Cartesian coordinates and generalises to unseen molecules. However, GeoDiff was not applied to proteins, but small molecules, and does not target the Boltzmann distribution. In contrast to Timewarp, [37] learn the transition probability for multiple time-resolutions, accurately capturing the dynamics. However, they do not show transferability between systems. Most similarly to Timewarp, in recent work, [8] trained a transferable ML model to simulate the time-coarsened dynamics of polymers. However, unlike Timewarp, their model acts on coarse grained representations. Additionally, it was not applied to proteins, and there is no MH step, which means that errors can accumulate in the simulation without being corrected.

*Markov state models* (MSMs) [32, 40, 10] work by running many short MD simulations, which are used to define a discrete state space, along with an estimated transition probability matrix. Similarly to Timewarp, MSMs estimate the transition probability between the state at a time $t$ and the time $t + \tau$, where $\tau \gg \Delta t$. Recent work has applied deep learning to MSMs, leading to *VAMPnets* [23] and *deep generative MSMs* [45], which replace the MSM data-processing pipeline with deep networks. In contrast to Timewarp, these models are not transferable and model the dynamics in a coarse-grained, discrete state space, rather than in the all-atom coordinate representation.

There has been much previous work on *neural adaptive samplers* [38, 20, 21], which use deep generative models as proposal distributions. *A-NICE-MC* [38] uses a volume-preserving flow trained using a likelihood-free adversarial method. Other methods use objective functions designed to encourage exploration. The entropy term in our objective function is inspired by [41]. In contrast to these methods, Timewarp focuses on *generalising* to new molecular systems without retraining.

Numerous enhanced sampling methods exist to for MD, such as parallel tempering [39, 6] or proposing updates of collective variables along transition paths [18, 27]. Given Timewarp's ability to accelerate MD, it often offers the opportunity to be integrated with these techniques.

## 3 Method

Consider the distribution of $x(t+\tau)$ induced by an MD simulation of Equation (2) for a time $\tau \gg \Delta t$, starting from $x(t)$. We denote this conditional distribution by $\mu(x(t + \tau)|x(t))$. Timewarp uses a deep probabilistic model to approximate $\mu(x(t + \tau)|x(t))$ (see Figure 1). Once trained, the model is used in an MCMC method to sample from the Boltzmann distribution.

### 3.1 Conditional normalising flows

We fit a *conditional normalising flow*, $p_\theta(x(t+\tau)|x(t))$, to $\mu(x(t + \tau)|x(t))$, where $\theta$ are learnable parameters. Normalising flows are defined by a base distribution (usually a standard Gaussian), and a *diffeomorphism* $f$, *i.e.* a differentiable bijection with a differentiable inverse. Specifically, we set $p_\theta(x(t+\tau)|x(t))$ as the density of the distribution defined by the following generative process:

$$z^p, z^v \sim \mathcal{N}(0, I), \quad x^p(t+\tau), x^v(t+\tau) \coloneqq f_\theta(z^p, z^v; x^p(t), x^v(t)). \tag{3}$$

---

[3]Please contact `andrewfoong@microsoft.com` for dataset access.

Here $z^p \in \mathbb{R}^{3N}$ and $z^v \in \mathbb{R}^{3N}$. For all settings of $\theta$ and $x(t)$, $f_\theta(\,\cdot\,; x(t))$ is a diffeomorphism that takes the latent variables $(z^p, z^v) \in \mathbb{R}^{6N}$ to $(x^p(t+\tau), x^v(t+\tau)) \in \mathbb{R}^{6N}$. The conditioning state $x(t)$ parameterises a family of diffeomorphisms, defining a *conditional* normalising flow [44]. Note that there are no invertibility constraints on the mapping from $x(t)$ to the output $x(t+\tau)$, only the map from $z$ to $x(t+\tau)$ must be invertible. Using the change of variables formula, we can evaluate:

$$p_\theta(x(t+\tau)|x(t)) = \mathcal{N}\left(f_\theta^{-1}(x(t+\tau); x(t)); 0, I\right) \left|\det \mathcal{J}_{f_\theta^{-1}(\cdot\,; x(t))}(x(t+\tau))\right|,$$

where $f_\theta^{-1}(\,\cdot\,; x(t)) : \mathbb{R}^{6N} \to \mathbb{R}^{6N}$ is the inverse of the diffeomorphism $f_\theta(\,\cdot\,; x(t))$, and $\mathcal{J}_{f_\theta^{-1}(\cdot\,; x(t))}(x(t+\tau))$ denotes the Jacobian of $f_\theta^{-1}(\,\cdot\,; x(t))$ evaluated at $x(t+\tau)$.

### 3.2 Dataset generation

We generate MD trajectories by integrating Equation (2) using the *OpenMM* library [7]. We simulate small proteins (peptides) in implicit water, *i.e.*, without explicitly modelling the degrees of freedom of the water molecules. Specifically, we generate a dataset of trajectories $\mathcal{D} = \{\mathcal{T}_i\}_{i=1}^P$, where $P$ is the number of peptides. Each MD trajectory is temporally subsampled with a spacing of $\tau$, so that $\mathcal{T}_i = (x(0), x(\tau), x(2\tau), \dots)$. During training, we randomly sample pairs $x(t), x(t+\tau)$ from $\mathcal{D}$. Each pair represents a sample from the conditional distribution $\mu(x(t+\tau)|x(t))$. Additional details are provided in Appendix E. Since the flow is trained on trajectory data from *multiple* peptides, we can deploy it at test time to generalise to *new* peptides not seen in the training data.

### 3.3 Augmented normalising flows

We are typically primarily interested in the distribution of the *positions* $x^p$, rather than the velocities $x^v$. Thus, it is not necessary for $x^v(t), x^v(t+\tau)$ to represent the actual velocities of the atoms in Equation (3). We hence simplify the learning problem by treating $x^v$ as *non-physical auxiliary variables* within the *augmented normalising flow* framework [9]. For each datapoint $x(t) = x^p(t), x^v(t)$ in $\mathcal{D}$, instead of obtaining $x^v(t)$ by recording the velocities in the MD trajectory, we *discard* the MD velocity and independently draw $x^v(t) \sim \mathcal{N}(0, I)$. The auxiliary variables $x^v(t)$ now contain no information about the future state $x^p(t+\tau), x^v(t+\tau)$, since $x^v(t)$ and $x^v(t+\tau)$ are drawn independently. Hence we can simplify $f_\theta$ to depend only on $x^p(t)$, with $x^p(t+\tau), x^v(t+\tau) := f_\theta(z^p, z^v; x^p(t))$. We include auxiliary variables for two reasons: First, they increase the expressivity of the distribution for $x^p$ without a prohibitive increase in computational cost [9, 2]. Second, constructing a conditional flow that respects *permutation equivariance* is simplified with auxiliary variables — see Section 4.1.

### 3.4 Targeting the Boltzmann distribution with asymptotically unbiased MCMC

After training the flow $p_\theta(x(t+\tau)|x(t))$, we use it as a proposal distribution in an MCMC method to target the joint distribution of the positions $x^p$ and the auxiliary variables $x^v$, which has density:

$$\mu_{\text{aug}}(x^p, x^v) \propto \exp\left(-\frac{U(x^p)}{k_B T}\right) \mathcal{N}(x^v; 0, I). \tag{4}$$

Starting from an initial state $X_0 = (X_0^p, X_0^v) \in \mathbb{R}^{6N}$ for state $m = 0$, we iterate:

$$\tilde{X}_m \sim p_\theta(\,\cdot\,|X_m^p), \quad X_{m+1} := \begin{cases} \tilde{X}_m & \text{with probability } \alpha(X_m, \tilde{X}_m) \\ X_m & \text{with probability } 1 - \alpha(X_m, \tilde{X}_m) \end{cases} \tag{5}$$

where $\alpha(X_m, \tilde{X}_m)$ is the *Metropolis-Hastings (MH) acceptance ratio* [24] targeting Equation (4):

$$\alpha(X, \tilde{X}) = \min\left(1, \frac{\mu_{\text{aug}}(\tilde{X}) p_\theta(X \mid \tilde{X}^p)}{\mu_{\text{aug}}(X) p_\theta(\tilde{X} \mid X^p)}\right) \tag{6}$$

The flow used for $p_\theta$ must allow for efficient sampling *and* exact likelihood evaluation, which is crucial for fast implementation of Equations (5) and (6). Additionally, after each MH step, we resample the auxiliary variables $X^v$ using a *Gibbs sampling* update:

$$(X_m^p, X_m^v) \leftarrow (X_m^p, \epsilon), \quad \epsilon \sim \mathcal{N}(0, I). \tag{7}$$

Iterating these updates yields a sample $X_m^p, X_m^v \sim \mu_{\text{aug}}$ as $m \to \infty$. To obtain a Boltzmann-distributed sample of the positions $X_m^p \sim \mu$, we simply discard the auxiliary variables $X_m^v$. As sending $m \to \infty$ is infeasible, we simulate the chain until a fixed budget is reached. In practice, we find that acceptance rates for our models can be low, around $1\%$. However, we stress that even with a low acceptance rate, our models can lead to faster exploration if the changes proposed are large enough, as we demonstrate in Section 6. Furthermore, we introduce a *batch sampling* procedure which significantly speeds up sampling whilst maintaining detailed balance. This procedure samples a batch of proposals with a single forward pass, and accepts the first proposal that meets the MH acceptance criterion. Pseudocode for the MCMC algorithm is given in Algorithm 1 in Appendix C.

### 3.5  Fast but biased exploration of the state space without MH corrections

Instead of using the MH correction to guarantee asymptotically unbiased samples, we can opt to use Timewarp in a simple *exploration* algorithm. In this case, we neglect the MH correction and accept all proposals with energy changes below some cutoff. This allows much faster exploration of the state space, and in Section 6 we show that, although technically biased, this often leads to qualitatively accurate free energy estimates. It also succeeds in discovering all metastable states orders of magnitude faster than Algorithm 1 and standard MD, which could be used, *e.g.*, to provide initialisation states for a subsequent MSM method. Pseudocode is given in Algorithm 2 in Appendix D.

## 4  Model architecture

We now describe the architecture of the flow $f_\theta(z^p, z^v; x^p(t))$, which is shown in Figure 2.

**RealNVP coupling flow**   Our architecture is based on RealNVP [5], which consists of a stack of *coupling layers* which affinely transform subsets of the dimensions of the latent variable based on the other dimensions. Specifically, we transform the position variables based on the auxiliary variables, and vice versa. In the $\ell$th coupling layer of the flow, the following transformations are implemented:

$$z_{\ell+1}^p = s_{\ell,\theta}^p(z_\ell^v; x^p(t)) \odot z_\ell^p + t_{\ell,\theta}^p(z_\ell^v; x^p(t)), \tag{8}$$

$$z_{\ell+1}^v = s_{\ell,\theta}^v(z_{\ell+1}^p; x^p(t)) \odot z_\ell^v + t_{\ell,\theta}^v(z_{\ell+1}^p; x^p(t)). \tag{9}$$

Going forward, we suppress the coupling layer index $\ell$. Here $\odot$ is the element-wise product, and $s_\theta^p : \mathbb{R}^{3N} \to \mathbb{R}^{3N}$ is our *atom transformer*, a neural network based on the transformer architecture [43] that takes the auxiliary latent variables $z^v$ and the conditioning state $x(t)$ and outputs scaling factors for the position latent variables $z^p$. The function $t_\theta^p : \mathbb{R}^{3N} \to \mathbb{R}^{3N}$ is implemented as another atom transformer, which uses $z^v$ and $x(t)$ to output a translation of the position latent variables $z^p$. The affine transformations of the position variables (in Equation (8)) are interleaved with similar affine transformations for the auxiliary variables (in Equation (9)). Since the scale and translation factors for the positions depend only on the auxiliary variables, and vice versa, the Jacobian of the transformation is lower triangular, allowing for efficient computation of the density. The full flow $f_\theta$ consists of $N_{\text{coupling}}$ stacked coupling layers, beginning from $z \sim \mathcal{N}(0, I)$ and ending with a sample from $p_\theta(x(t+\tau)|x(t))$. This is depicted in Figure 2, Left. Note that there is a skip connection such that the output of the flow predicts the *change* $x(t+\tau) - x(t)$, rather than $x(t+\tau)$ directly.

**Atom transformer**   We now describe the *atom transformer* network. Let $x_i^p(t), z_i^p, z_i^v$, all elements of $\mathbb{R}^3$, denote respectively the position of atom $i$ in the conditioning state, the position latent variable for atom $i$, and the auxiliary latent variable for atom $i$. To implement an atom transformer which takes $z^v$ as input (such as $s_\theta^p(z^v, x^p(t))$ and $t_\theta^p(z^v, x^p(t))$ in Equation (8)), we first concatenate the variables associated with atom $i$. This leads to a vector $a_i := [x_i^p(t), h_i, z_i^v]^\top \in \mathbb{R}^{H+6}$, where $z_i^p$ has been excluded since $s_\theta^p, t_\theta^p$ are not allowed to depend on $z^p$. Here $h_i \in \mathbb{R}^H$ is a learned embedding vector which depends only on the atom type. The vectors $a_i$ are fed into an MLP $\phi_{\text{in}} : \mathbb{R}^{H+6} \to \mathbb{R}^D$, where $D$ is the feature dimension of the transformer. The vectors $\phi_{\text{in}}(a_1), \ldots, \phi_{\text{in}}(a_N)$ are then fed into $N_{\text{transformer}}$ stacked transformer layers. After the transformer layers, they are passed through another atom-wise MLP, $\phi_{\text{out}} : \mathbb{R}^D \to \mathbb{R}^3$. The final output is in $\mathbb{R}^{3N}$ as required. This is depicted in Figure 2, Middle. When implementing $s_\theta^v$ and $t_\theta^v$ from Equation (9), a similar procedure is performed on the vector $[x_i^p(t), h_i, z_i^p]^\top$, but now including $z_i^p$ and excluding $z_i^v$. There are two key differences between the atom transformer and the architecture in [43]. First, to maintain permutation equivariance,

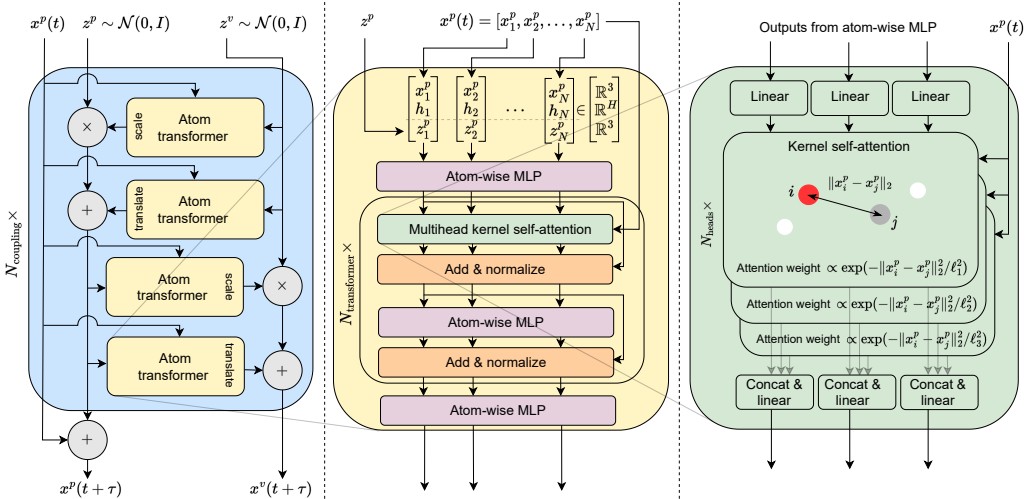

Figure 2: Schematic illustration of the Timewarp conditional flow architecture, described in Section 4. *Left*: A single conditional RealNVP coupling layer. *Middle*: A single atom transformer module. *Right*: the multihead kernel self-attention module.

we do not use a positional encoding. Second, instead of dot product attention, we use a simple *kernel self-attention* module, which we describe next.

**Kernel self-attention**  We motivate the kernel self-attention module with the observation that physical forces acting on the atoms in a molecule are *local*: *i.e.*, they act more strongly on nearby atoms. Intuitively, for values of $\tau$ that are not too large, the positions at time $t + \tau$ will be more influenced by atoms that are nearby at time $t$, compared to atoms that are far away. Thus, we define the attention weight $w_{ij}$ for atom $i$ attending to atom $j$ as follows:

$$w_{ij} = \frac{\exp(-\|x_i^p - x_j^p\|_2^2/\ell^2)}{\sum_{j'=1}^{N} \exp(-\|x_i^p - x_{j'}^p\|_2^2/\ell^2)}, \tag{10}$$

where $\ell$ is a lengthscale parameter. The outputs $\{r_{\text{out},i}\}_{i=1}^{N}$, given the inputs $\{r_{\text{in},i}\}_{i=1}^{N}$, are then:

$$r_{\text{out},i} = \sum_{j=1}^{N} w_{ij} V \cdot r_{\text{in},j}, \tag{11}$$

where $V \in \mathbb{R}^{d_{\text{out}} \times d_{\text{in}}}$ is a learnable matrix. This kernel self-attention is an instance of the RBF kernel attention investigated in [42]. Similarly to [43], we introduce a *multihead* version, where each head has a different lengthscale. This is illustrated in Figure 2, Right. We found that kernel self-attention was significantly faster to compute than dot product attention, and performed similarly.

## 4.1 Symmetries

The MD dynamics respects certain physical *symmetries* that would be advantageous to incorporate. We now describe how each of these symmetries is incorporated in Timewarp.

**Permutation equivariance**  Let $\sigma$ be a permutation of the $N$ atoms. Since the atoms have no intrinsic ordering, the only effect of a permutation of $x(t)$ on the future state $x(t + \tau)$ is to permute the atoms similarly, *i.e.*,

$$\mu(\sigma x(t + \tau)|\sigma x(t)) = \mu(x(t + \tau)|x(t)). \tag{12}$$

Our conditional flow satisfies permutation equivariance exactly. To show this, we use the following proposition proved in Appendix A.1, which is an extension of [16, 36] for conditional flows:

**Proposition 4.1.** *Let $\sigma$ be a symmetry action, and let $f(\,\cdot\,;\,\cdot\,)$ be an equivariant map such that $f(\sigma z; \sigma x) = \sigma f(z; x)$ for all $\sigma, z, x$. Further, let the base distribution $p(z)$ satisfy $p(\sigma z) = p(z)$ for all $\sigma, z$. Then the conditional flow defined by $z \sim p(z)$, $x(t + \tau) := f(z; x(t))$ satisfies $p(\sigma x(t + \tau)|\sigma x(t)) = p(x(t + \tau)|x(t))$.*

Our flow satisfies $f_\theta(\sigma z; \sigma x(t)) = \sigma f_\theta(z; x(t))$ since the transformer is permutation equivariant, and permuting $z$ and $x(t)$ together permutes the inputs. Furthermore, the base distribution $p(z) = \mathcal{N}(0, I)$ is permutation invariant. Note that the presence of auxiliary variables allows us to easily construct a permutation equivariant coupling layer.

**Translation and rotation equivariance**  Consider a transformation $T = (R, a)$ that acts on $x^p$:

$$T x_i^p = R x_i^p + a, \quad 1 \le i \le N, \tag{13}$$

where $R$ is a $3 \times 3$ rotation matrix, and $a \in \mathbb{R}^3$ is a translation vector. We would like the model to satisfy $p_\theta(Tx(t+\tau)|Tx(t)) = p_\theta(x(t+\tau)|x(t))$. We achieve translation equivariance by subtracting the average position of the atoms in the initial state (Appendix A.2). Rotation equivariance is not encoded in the architecture but is handled by data augmentation: each training pair $(x(t), x(t + \tau))$ is acted upon by a random rotation matrix $R$ to form $(Rx(t), Rx(t + \tau))$ in each iteration.

## 5  Training objective

The model is trained in two stages: (i) *likelihood training*, the model is trained via maximum likelihood on pairs of states from the trajectories in the dataset. Let $k$ index training pairs, such that $\{(x^{(k)}(t), x^{(k)}(t + \tau))\}_{k=1}^{K}$ represents all pairs of states at times $\tau$ apart in $\mathcal{D}$. We optimise:

$$\mathcal{L}_{\mathrm{lik}}(\theta) := \tfrac{1}{K} \textstyle\sum_{k=1}^{K} \log p_\theta(x^{(k)}(t + \tau)|x^{(k)}(t)). \tag{14}$$

(ii) *acceptance training*, the model is fine-tuned to maximise the probability of MH acceptance. Let $x^{(k)}(t)$ be sampled uniformly from $\mathcal{D}$. Then, we use the model to sample $\tilde{x}_\theta^{(k)}(t+\tau) \sim p_\theta(\cdot|x^{(k)}(t))$ using Equation (3). We use this to optimise the acceptance probability in Equation (6) with respect to $\theta$. Let $r_\theta(X, \tilde{X})$ denote the model-dependent term in the acceptance ratio in Equation (6):

$$r_\theta(X, \tilde{X}) := \frac{\mu_{\mathrm{aug}}(\tilde{X}) p_\theta(X \mid \tilde{X}^p)}{\mu_{\mathrm{aug}}(X) p_\theta(\tilde{X} \mid X^p)}. \tag{15}$$

The acceptance objective is then given by:

$$\mathcal{L}_{\mathrm{acc}}(\theta) := \tfrac{1}{K} \textstyle\sum_{k=1}^{K} \log r_\theta(x^{(k)}(t), \tilde{x}_\theta^{(k)}(t + \tau)). \tag{16}$$

Training to maximise the acceptance probability can lead to the model proposing changes that are too small: if $\tilde{x}_\theta^{(k)}(t + \tau) = x^{(k)}(t)$, then all proposals will be accepted. To mitigate this, during acceptance training, we use an objective which is a weighted average of $\mathcal{L}_{\mathrm{acc}}(\theta)$, $\mathcal{L}_{\mathrm{lik}}(\theta)$ and a Monte Carlo estimate of the average differential entropy,

$$\mathcal{L}_{\mathrm{ent}}(\theta) := -\tfrac{1}{K} \textstyle\sum_{k=1}^{K} \log p_\theta(\tilde{x}_\theta^{(k)}(t + \tau)|x^{(k)}(t)). \tag{17}$$

## 6  Experiments

We evaluate Timewarp on small peptide systems. To compare with MD, we focus on the slowest transitions between metastable states, as these are the most difficult to traverse. To find these, we use *time-lagged independent component analysis* (TICA) [30], a linear dimensionality reduction technique that maximises the autocorrelation of the transformed coordinates. The slowest components, TIC 0 and TIC 1, are of particular interest. To measure the speed-up achieved by Timewarp, we compute the *effective sample size* per second of wall-clock time (ESS/s) for the TICA components. The ESS/s is given by

$$\mathrm{ESS/s} = \frac{M_{\mathrm{eff}}}{t_{\mathrm{sampling}}} = \frac{M}{t_{\mathrm{sampling}} \left(1 + 2 \sum_{\tau=1}^{\infty} \rho_\tau\right)}, \tag{18}$$

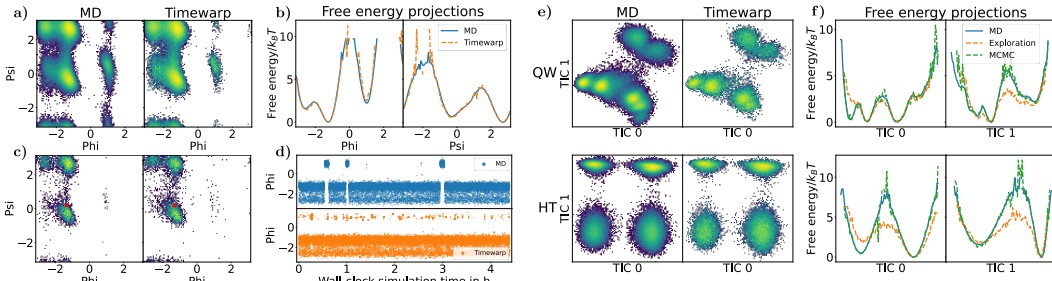

Figure 3: *Left half:* **Alanine dipeptide experiments.** (a) Ramachandran plots for MD and Timewarp samples generated according to Algorithm 1. (b) Free energy comparison for the two dihedral angles $\varphi$ and $\psi$. (c) Ramachandran plots for the conditional distribution of MD compared with the Timewarp model. Red cross denotes initial state. (d) Time dependence of the $\varphi$ dihedral angle of MD and the Markov chain generated with the Timewarp model. *Right half:* **Experiments on 2AA test dipeptides QW (top row) and HT (bottom row).** (e) TICA plots for a long MD chain and samples generated with the Timewarp MCMC algorithm (Algorithm 1). (f) Free energy comparison for the MD trajectory, Timewarp MCMC (Algorithm 1), and Timewarp exploration (Algorithm 2).

where $M$ is the chain length, $M_{\text{eff}}$ is the effective number of samples, $t_{\text{sampling}}$ is the sampling wall-clock time, and $\rho_\tau$ is the autocorrelation for the lag time $\tau$ [26]. The speed-up factor is defined as the ESS/s achieved by Timewarp divided by the ESS/s achieved by MD. Additional experiments and results can be found in Appendix B. We train three flow models on three datasets: (i) *AD*, consisting of simulations of alanine dipeptide, (ii) *2AA*, with peptides with 2 amino acids, and (iii) *4AA*, with peptides with 4 amino acids. All datasets are created with MD simulations performed with the same parameters (see Appendix E). For 2AA and 4AA, we train on a randomly selected trainset of short trajectories ($50\text{ns} = 10^8$ steps), and evaluate on unseen test peptides. The relative frequencies of the amino acids in 2AA and 4AA are similar across the splits. For 4AA, the training set consists of about $1\%$ of the total number of possible tetrapeptides ($20^4$), making the generalisation task significantly more difficult than for 2AA. For more details see Table 2 in Appendix E.

**Alanine dipeptide (AD)**  We first investigate alanine dipeptide, a small (22 atoms) single peptide molecule. We train Timewarp on AD as described in Section 5 and sample new states using Algorithm 1 for a chain length of 10 million, accepting roughly $2\%$ of the proposals. In Figure 3a we visualise the samples using a *Ramachandran plot* [33], which shows the distribution of the backbone dihedral angles $\varphi$ and $\psi$. Each mode in the plot represents a metastable state. We see that the Timewarp samples closely match MD, visiting all the metastable states with the correct relative weights. In Figure 3b we plot the free energy (*i.e.*, the relative log probability) of the $\varphi$ and $\psi$ angles, again showing close agreement. The roughness in the plot is due to some regions of state space having very few samples. In Figure 3c we show, for an initial state $x(t)$, the conditional distribution of MD obtained by integrating Equation (2), $\mu(x(t+\tau)|x(t))$, compared with the model $p_\theta(x(t+\tau)|x(t))$, demonstrating close agreement. Finally, Figure 3d shows the time-evolution of the $\varphi$ angle for MD and Timewarp. Timewarp exhibits significantly more transitions between the metastable states than MD. As a result, the autocorrelation along the $\varphi$ angle decays much faster in terms of wall-clock time, resulting in a $\approx 7\times$ speed-up in terms of ESS/s compared to MD (see Appendix B.4).

**Dipeptides (2AA)**  Next, we demonstrate transferability on dipeptides in 2AA. After training on the train dipeptides, we deploy Timewarp with Algorithm 1 on the test dipeptides for a chain length of 20 million. Timewarp achieves acceptance probabilities between $0.03\%$ and $2\%$ and explores all metastable states (Appendix B.1). The results are shown for the dipeptides QW and HT in Figure 3ef, showing close agreement between Timewarp and long MD chains ($1\ \mu s = 2 \times 10^9$ steps). For these dipeptides, Timewarp achieves ESS/s speed-up factors over MD of 5 and 33 respectively (Appendix B.4). In Figure 4, Left, we show the speed-up factors for Timewarp verses MD for each of the 100 test dipeptides. Timewarp provides a median speed-up factor of about five across these peptides. In addition, we generate samples with the Timewarp model *without* the MH correction as detailed in Section 3.5. We sample 100 parallel chains for only 10000 steps starting from the same

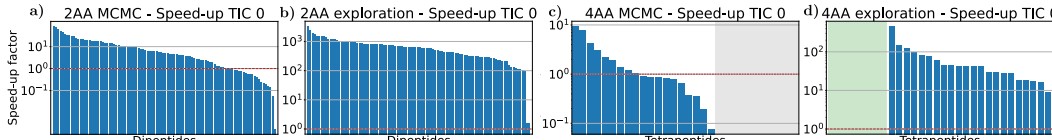

Figure 4: Speed-up factors in terms of ESS/s ratios for the slowest TICA component for the Timewarp MCMC and exploration algorithms, compared to MD. The dashed red line shows a speed-up factor of one. Gray areas depict peptides where Timewarp fails to explore all meta-stable states within 20 million steps, but MD does. Green areas depict peptides where MD fails to find all metastable states, but Timewarp does. (a), (c) Speed-up for the Timewarp MCMC algorithm (Algorithm 1) on test dipeptides (2AA) and tetrapeptides (4AA), respectively. (b), (d) Speed-up for the Timewarp exploration algorithm (Algorithm 2) on test dipeptides (2AA) and tetrapeptides (4AA), respectively.

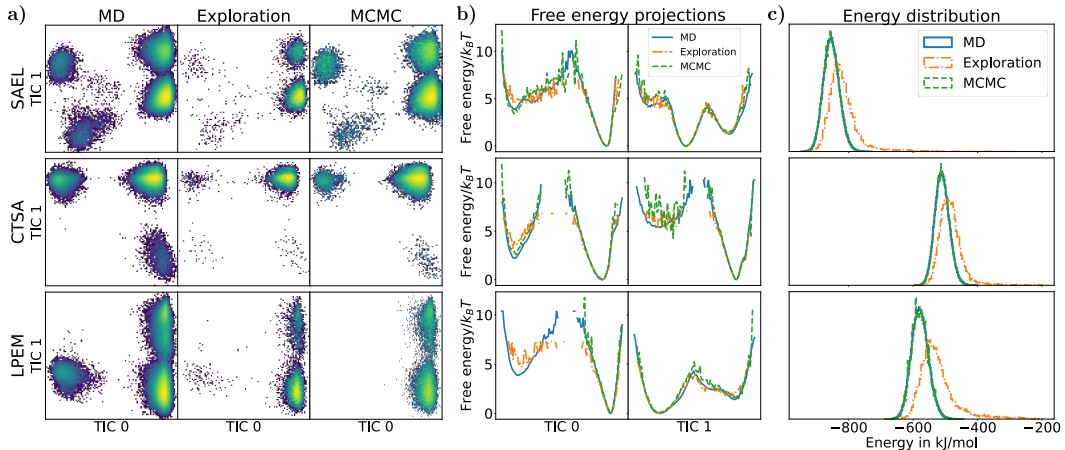

Figure 5: Experiments on 4AA test tetrapeptides SAEL, CTSA and LPEM (top, middle and bottom rows respectively). Samples were generated via MD, Timewarp exploration (Algorithm 2), and Timewarp MCMC (Algorithm 1). (a) TICA plots of samples. (b) Free energies along the first two TICA components. (c) Potential energy distribution.

initial state for each test peptide. For each peptide we select *only one* of these chains that finds all meta-stable states for evaluations. As before, we compute the ESS/s to compare with MD, showing a median speedup factor of $\approx 600$ (Figure 4c). Note that the actual speedup when using all the chains sampled in parallel will be much larger. Timewarp exploration leads to free energy estimates that are qualitatively similar to MD, but less accurate than Timewarp MCMC (Figure 3f).

**Tetrapeptides (4AA)** Finally, we study the more challenging 4AA dataset. After training on the trainset, we sample 20 million Markov chain states for each test tetrapeptide using Algorithm 1 and compare with long MD trajectories ($1\mu s$). In contrast to the simpler dipeptides, both Timewarp MCMC and the long MD trajectories miss some metastable states. However, Timewarp in exploration mode (Algorithm 2) can be used as a validation tool to quickly verify exploration of the whole state space. Figure 5a shows that metastable states unexplored by MD and Timewarp MCMC can be found by the Timewarp exploration algorithm. We carefully confirm the physical validity of these discovered states by running shorter MD trajectories in their vicinity (see Appendix B.5), to ensure that they are not simply artefacts invented by the model. As with 2AA, we again report the speedup factors for Timewarp relative to MD in Figure 4b,d. Although Timewarp MCMC fails to speed up sampling for most tetrapeptides, Timewarp *exploration* shows a median speedup factor of $\approx 50$. For 8 test tetrapeptides, MD fails to explore all metastable states, whereas Timewarp succeeds — these are marked in green. For 10 tetrapeptides, Timewarp MCMC fails to find all metastable states found by MD — these are marked in grey. Figure 5b shows that when Timewarp MCMC discovers all metastable states, its free energy estimates match those of MD very well. However, it sometimes

misses metastable states leading to poor free energy estimates in those regions. Figure 5c shows that Timewarp MCMC also leads to a potential energy distribution that matches MD very closely. In contrast, Timewarp exploration discovers all metastable states (even ones that MD misses), but has less accurate free energy plots. It also has a potential energy distribution that is slightly too high relative to MD and Timewarp MCMC.

## 7    Limitations / Future work

The Timewarp MCMC algorithm generates low acceptance probabilities ($< 1\%$) for most peptides (see Appendix B.1). However, this is not a limitation in itself. In general, a larger proposal timestep $\tau$ yields smaller acceptance rates as the prediction problem becomes more difficult. However, due to Algorithm 1, we can evaluate multiple samples in parallel at nearly no additional costs. As a result, a lower acceptance rate, when coupled with a larger timestep $\tau$, is often a favorable trade-off. While we speed-up only roughly a third of the 4AA peptides when using the MH correction, beating MD in wall-clock time on unseen peptides in the all-atom representation is a challenging task which has not been demonstrated by ML methods before. Furthermore, one could consider targeting systems using a semi-empirical force field instead of a classical one. Given that Timewarp requires considerably fewer energy evaluations than MD simulations, one can anticipate a more substantial acceleration in this context.

Although MD and Timewarp MCMC fail to find some metastable states that were found by Timewarp exploration, we refrained from running MD and Timewarp MCMC longer due to the high computational cost (Appendix F). Timewarp generates fewer samples compared to traditional MD simulations within the same timeframe. Consequently, this scarcity of samples becomes even more pronounced in transition states, which makes Timewarp difficult to apply to chemical reactions.

Timewarp could be integrated with other enhanced sampling methods, like parallel tempering or transition path sampling. In the case of parallel tempering, the effective integration requires the training of the Timewarp model across multiple temperatures, which then allows to sample all the replicas with Timewarp instead of MD. We could also alternate Timewarp proposals with learned updates to collective variables, like dihedral angles. These combined steps would still allow unbiased sampling from the target distribution [27].

Moreover, we only studied small peptide systems in this work. Scaling Timewarp to larger systems remains a topic for future research, and there are several promising avenues to consider. One approach is to explore different network architectures, potentially capturing all the symmetries inherent in the system. Another option is to study coarse-grained structures instead of all-atom representations, to reduce the dimensionality of the problem.

## 8    Conclusion

We presented Timewarp, a transferable enhanced sampling method which uses deep networks to propose large conformational changes when simulating molecular systems. We showed that Timewarp used with an MH correction can accelerate asymptotically unbiased sampling on many unseen dipeptides, allowing faster computation of equilibrium expectation values. Although this acceleration was only possible for a minority of the *tetrapeptides* we considered, we showed that Timewarp used *without* the MH correction explores the metastable states of both dipeptides and tetrapeptides much faster than standard MD, and we verify the metastable states discovered are physically meaningful. This provides a promising method to quickly validate if MD simulations have visited all metastable states. Although further work needs to be done to scale Timewarp to larger, more interesting biomolecules, this work clearly demonstrates the ability of deep learning algorithms to leverage transferability to accelerate the MD sampling problem.

## Acknowledgments

We thank Bas Veeling, Claudio Zeni, Andrew Fowler, Lixin Sun, Chris Bishop, Rianne van den Berg, Hannes Schulz, Max Welling and the entire Microsoft AI4Science team for insightful discussions and computing help.

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

## A    Symmetries of the architecture

### A.1    Proof of Proposition 4.1

In this appendix we provide more details on the equivariance of the Timewarp architecture. We first prove Proposition 4.1 from the main body:

*Proof.* Let $X(t+\tau)_{x(t)}$ denote the random variable obtained by sampling $Z \sim p(z)$ and computing $X(t+\tau) := f(Z; x(t))$. Here we subscript $X(t+\tau)_{x(t)}$ by $x(t)$ to emphasize that this is the random variable obtained when conditioning the flow on $x(t)$. We first note that the equivariance condition on the densities $p(\sigma x(t+\tau)|\sigma x(t)) = p(x(t+\tau)|x(t))$ is equivalent to the following constraint on the random variables:

$$X(t+\tau)_{\sigma x(t)} \stackrel{d}{=} \sigma X(t+\tau)_{x(t)}, \tag{19}$$

where $\stackrel{d}{=}$ denotes equality in distribution. To see this, let $p_X$ denote the density of the random variable $X$. Then, in terms of densities, Equation (19) is equivalent to stating that, for all $x$,

$$p_{X(t+\tau)_{\sigma x(t)}}(x) = p_{\sigma X(t+\tau)_{x(t)}}(x) \tag{20}$$

$$= p_{X(t+\tau)_{x(t)}}(\sigma^{-1}x), \tag{21}$$

where in Equation (21) we used the change-of-variables formula, along with the fact that the group actions we consider (rotations, translations, permutations) have unit absolute Jacobian determinant. Redefining $x \leftarrow \sigma x$, we get that for all $x$,

$$p_{X(t+\tau)_{\sigma x(t)}}(\sigma x) = p_{X(t+\tau)_{x(t)}}(x). \tag{22}$$

Recalling the notation that $X(t+\tau)_{x(t)}$ is interpreted as the random variable obtained by conditioning the flow on $x(t)$, this can be written as

$$p(\sigma x|\sigma x(t)) = p(x|x(t)) \tag{23}$$

which is exactly the equivariance condition stated in terms of densities above. Having rephrased the equivariance condition in terms of random variables in Equation (19), the proof of Proposition 4.1 is straightforward.

$$X(t+\tau)_{\sigma x(t)} := f(Z, \sigma x(t)) \tag{24}$$

$$\stackrel{d}{=} f(\sigma Z, \sigma x(t)) \tag{25}$$

$$= \sigma f(Z, x(t)) \tag{26}$$

$$:= \sigma X(t+\tau)_{x(t)}, \tag{27}$$

where in Equation (25) we used the fact that the base distribution $p(z)$ is $\sigma$-invariant. $\square$

### A.2    Translation equivariance via canonicalisation

We now describe the canonicalisation technique used to make our models translation equivariant. Let $q(x^p(t+\tau), x^v(t+\tau)|x^p(t))$ be an arbitrary conditional density model, which is not necessarily translation equivariant. We can make it translation equivariant in the following way. Let $\overline{x^p}$ denote the average position of the atoms,

$$\overline{x^p} := \frac{1}{N}\sum_{i=1}^{N} x_i^p. \tag{28}$$

Then we define

$$p(x^p(t+\tau), x^v(t+\tau)|x^p(t)) := q(x^p(t+\tau) - \overline{x^p(t)}, x^v(t+\tau)|x^p(t) - \overline{x^p(t)}) \tag{29}$$

where the subtraction of $\overline{x^p(t)}$ is broadcasted over all atoms. We now consider the effect of translating both $x^p(t)$ and $x^p(t+\tau)$ by the same amount. Let $a$ be a translation vector in $\mathbb{R}^3$. Then

$$p(x^p(t+\tau) + a, x^v(t+\tau)|x^p(t) + a) \tag{30}$$

$$= q(x^p(t+\tau) + a - \overline{(x^p(t) + a)}, x^v(t+\tau)|x^p(t) + a - \overline{(x^p(t) + a)}) \tag{31}$$

$$= q(x^p(t+\tau) + a - \overline{x^p(t)} - a, x^v(t+\tau)|x^p(t) + a - \overline{x^p(t)} - a) \tag{32}$$

$$= q(x^p(t+\tau) - \overline{x^p(t)}, x^v(t+\tau)|x^p(t) - \overline{x^p(t)}) \tag{33}$$

$$= p(x^p(t+\tau), x^v(t+\tau)|x^p(t)). \tag{34}$$

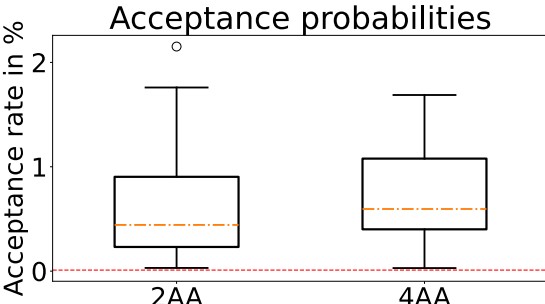

Figure 6: Acceptance probabilities for samples on unseen test peptides with the Timewarp MCMC algortihm. The red line is at $0.01\%$, below that efficient sampling becomes difficult.

Hence $p$ is translation equivariant even if $q$ is not.

### A.3 Chirality

In addition to the symmetries described in Section 4.1 the potential energy $U(x^p)$ of a molecular configuration is also invariant under mirroring. However, in the presence of *chirality* centers, a mirrored configuration is non-superposable to its original image [12]. An example of a chirality center in an amino acid is a Carbon atom connected to four different groups, e.g. a $C_\alpha$ atom. In nature most amino acids come in one form, namely L-amino acids. Hence, all our datasets consist of peptides containing only L-amino acids. In rare cases, as the model proposes large steps, one step might change one L-amino acid of a peptide to a D-amino acid in a way that the resulting configuration has a low energy and the step would be accepted. We prevent this by checking all chirality centers for changes at each step and reject samples where such a change occurs. This does not add any significant computational overhead.

## B Additional results

In this section we show additional results like the conditional distribution as well as more peptide examples for experiments discussed in Section 6.

### B.1 2AA additional results

More examples from the 2AA test set are presented in Figures 7 and 8. We achieve the worst speed-up for the dipeptide GP ( Figure 7 last row) as it does not show any slow transitions. The distribution of the acceptance probabilities for the Timewarp MCMC algorithm is shown in Figure 6

### B.2 4AA additional results

More examples from the 4AA test set are presented in Figures 9 and 11. The distribution of the acceptance probabilities for the Timewarp MCMC algorithm is shown in Figure 6.

### B.3 Conditional distributions

The model was trained to generate samples from the conditional Boltzmann distribution $\mu(x(t + \tau)|x(t))$. Here we show some examples of the conditional distribution generated by the Timewarp model compared to the conditional distribution induced by MD. While we can generate $5,000$ samples from the conditional distribution of the model in parallel, we require $5,000$ distinct MD trajectories of simulation length $\tau$ to sample the conditional distribution with MD. Hence, generating samples from the conditional distribution is several orders of magnitude faster with the model. In Figures 10 to 12 we show example conditional distributions for alanine dipeptide and peptides from the 2AA and 4AA datasets. For all peptides the model learns a conditional distribution that is close to the conditional MD distribution. Moreover, the relative weights in the TICA projections as well as the bondlength distributions match very well. Only the energies of the model samples are higher,

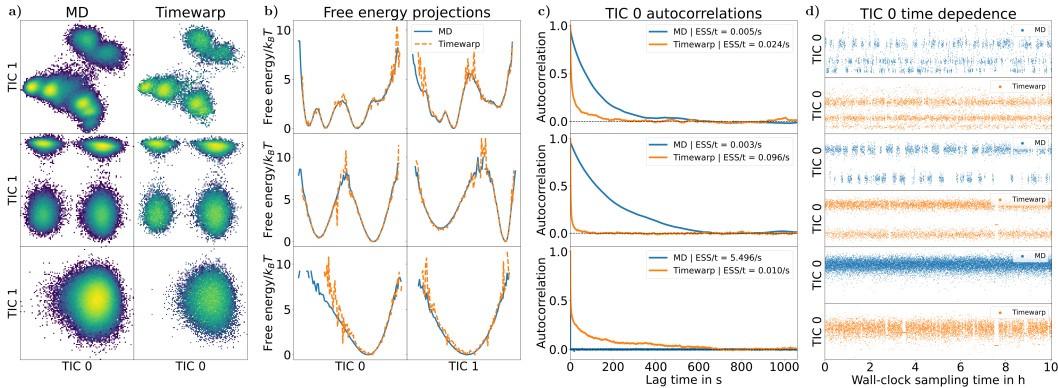

Figure 7: Experiments on 2AA test dipeptides QW (top row), HT (middle row) and GP (bottom row). Comparison of the long MD trajectory and Timewarp MCMC (Algorithm 1). (a) TICA plots. (b) Free energy comparison of the first two TICA components. (c) Autocorrelation for the TIC 0 component. (d) Time dependence of the TIC 0 component.

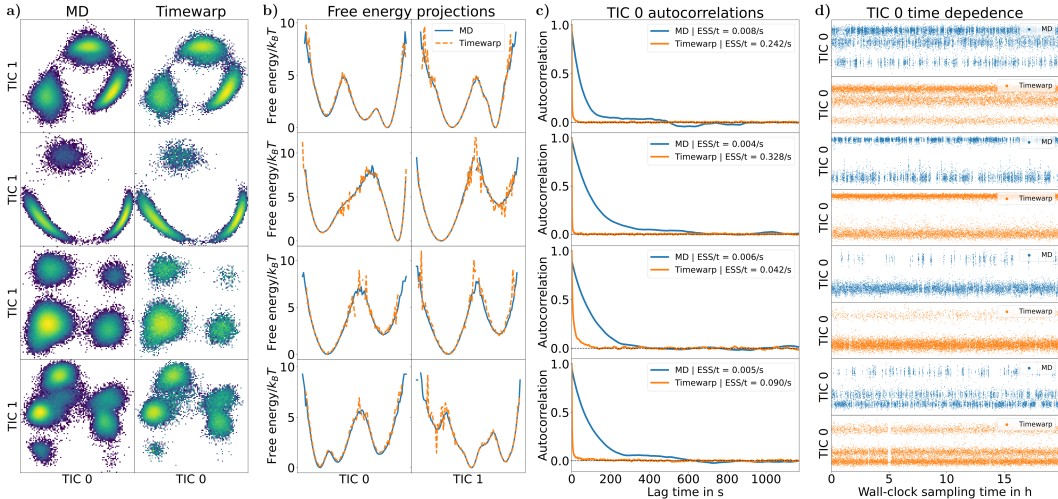

Figure 8: Experiments for the 2AA test dipeptides DH (first row), GT (second row), TK (third row), and CW (last row). Comparison of the long MD trajectory and Timewarp MCMC (Algorithm 1). (a) TICA plots. (b) Free energy comparison of the first two TICA components. (c) Autocorrelation for the TIC 0 component. (d) Time dependence of the TIC 0 component.

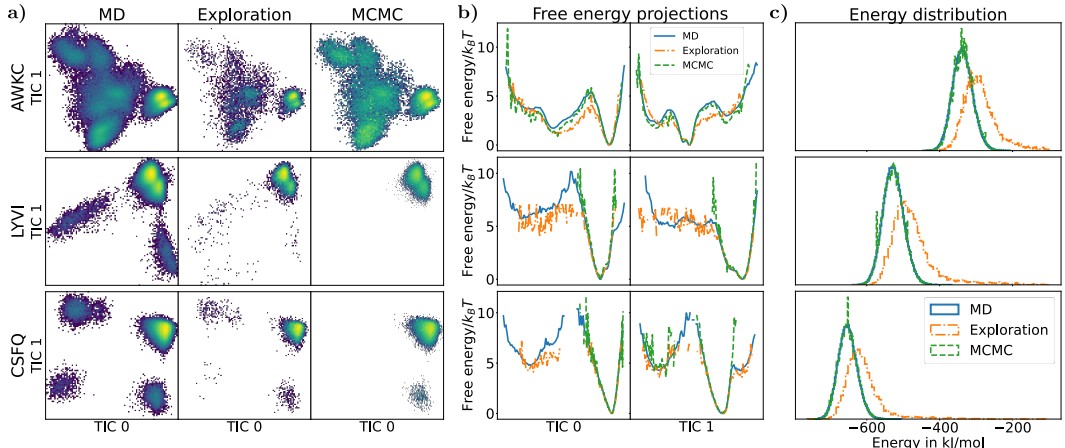

Figure 9: Experiments on 4AA test tetrapeptides AWCK, LYVI and CSFQ (top, middle and bottom rows respectively). Samples were generated via MD, Timewarp exploration (Algorithm 2), and Timewarp MCMC (Algorithm 1). (a) TICA plots of samples. (b) Free energies along the first two TICA components. (c) Potential energy distribution. For AWCK all metastable states are found by all methods, for LYVI the MD trajectory misses one state, and for CSFQ Timewarp MCMC misses the slowest transition. In all cases Timewarp exploration discovers all metastable states.

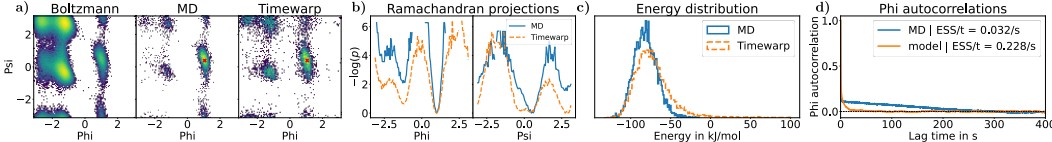

Figure 10: Comparing the conditional Boltzmann distributions generated with MD trajectories and the Timewarp model for alanine dipeptide. (a) Ramachandran plots for the conditional distributions compared with the equilibrium Boltzmann distribution. The red cross indicates the conditioning state. This is similar to the plot shown in Figure 3c, but here showing a different conditioning state. The match between the conditional distributions is not as close here as it is for Figure 3c, which could be because here the conditioning state is chosen to be in the less likely metastable state. (b) Projections on the first two dihehdral angles for the conditional distributions. (c) Potential energies of the conditional distributions. (d) Autocorrelations for samples generated according to the MCMC algorithm (Algorithm 1) compared with a long MD trajectory. Note that this autocorrelation plot is not for the conditional distribution, but corresponds to the results shown in Figure 3.

emphasising the importance of the Metropolis-Hastings correction to obtain unbiased samples from the Boltzmann distribution with the Timewarp model.

### B.4   Autocorrelations

In Section 6 we compute the speedup of the Timewarp model by comparing the effective sample sizes per second (Equation (18)) for the slowest transition with MD. As the ESS depends on the autocorrelation, it is also insightful to look at the autocorrelation decay in terms of wall-clock time. We show some example autocorrelations for the investigated peptides in Figures 7, 8 and 10. Note that the area under the autocorrleation curve is inversely proportional to the ESS.

### B.5   Exploration of new metastable states

For some tetrapetides in the test set even long MD trajectories ($1\mu$s) miss some metastable states, *e.g.* for LYVI and CSTA shown in Figure 12a. However, we can easily explore these with the Timewarp exploration algorithm (Algorithm 2). To confirm that these additional metastable states are in fact

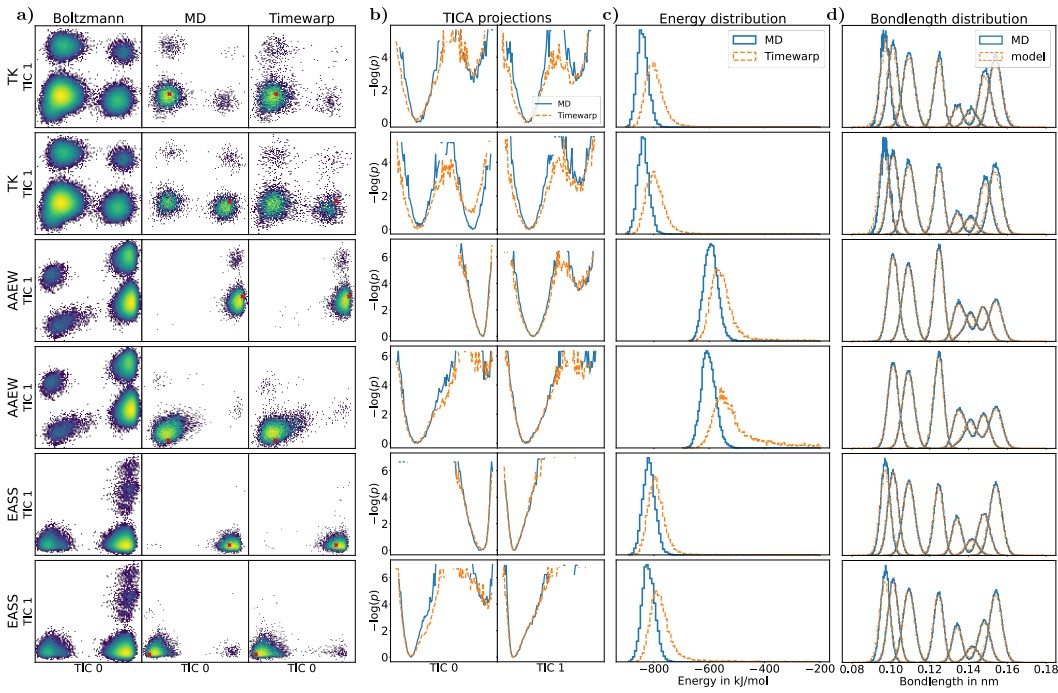

Figure 11: Comparing the conditional distribution of the Timewarp model $p_\theta(x(t+\tau)|x(t))$ with the conditional distribution generated with MD $\mu(x(t+\tau)|x(t))$. The rows correspond to the peptides TK, AAEW, EASS, respectively, where we show for each peptide two different conditioning states. (a) TICA plots. First column: samples from the Boltzmann distribution $\mu(x)$ generated with MD. Second column: samples from $\mu(x(t+\tau)|x(t))$ generated with MD. The conditioning state $x(t)$ is indicated with the red cross. Third column: samples from $p_\theta(x(t+\tau)|x(t))$ generated with the Timewarp conditional flow, without MH correction. The conditioning state $x(t)$ is indicated with the red cross. (b) Projection of the conditional distributions from (a) onto the first two TICA components. (c) Potential energy distributions of the conditional distributions. (d) Conditional bondlength distribution, which for these values of $\tau$ will be close to the equilibrium distribution. Each mode in the graph represents a different bond type, $e.g.$, C-H.

stable, we run several shorter MD trajectories ($0.5 \times 10^6$fs), in the same way as in Appendix B.3, starting in a nearby metastable state already discovered with the long MD trajectory (Figure 12b). Once one of them hits the new, previously undiscovered state, we start new short MD trajectories ($0.5 \times 10^6$fs) from there as well (Figure 12c). These new short MD trajectories either sample within this previously undiscovered state, or transition to the other metastable states. This shows that this metastable state discovered by Timewarp exploration is indeed valid, and was simply undiscovered during the long MD trajectory. In addition, we compare in Figure 12 the conditional MD distributions with that of Timewarp, again showing close agreement.

## C   Batched sampling algorithm for Timewarp with MH corrections

Pseudocode for the algorithm described in Section 3.4 is given in Algorithm 1.

## D   Exploration of metastable states using Timewarp without MH corrections

We describe the exploration algorithm for Timewarp, which accepts all proposed states unless the energy is above a certain cutoff value. As there is no MH correction, the generated samples will not asymptotically follow the Boltzmann distribution, but the exploration of the state space is much faster than with MD or Algorithm 1. The pseudocode is shown in Algorithm 2:

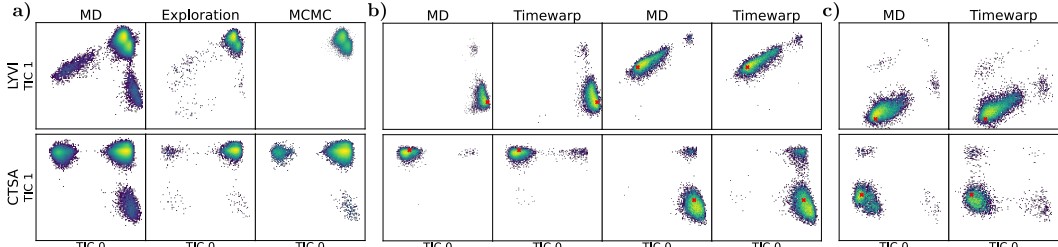

Figure 12: Validation of new metastable states found with Timewarp exploration. The red crosses indicate the different conditioning states. (a) TICA plots for (not conditional) samples generated via a long MD trajectory, Timewarp exploration (Algorithm 2), and Timewarp MCMC (Algorithm 1). Timewarp exploration discovers some metastable states unseen in the long MD trajectories (bottom of LYVI plot, bottom left of CTSA plot). (b) Conditional distributions generated with MD or with the Timewarp model, starting from states visited by the *long* MD trajectory shown in (a). Some short MD trajectories now discover the new metastable states, verifying that they are indeed valid states. Only the final state of each short MD trajectory is recorded. (c) Conditional distributions generated with MD or with the Timewarp model, starting from the new metastable states discovered by one of the *short* MD trajectories shown in (b).

---

**Algorithm 1** Timewarp MH-corrected MCMC with batched proposals

---

**Require:** Initial state $X_0 = (X_0^p, X_0^v)$, chain length $M$, proposal batch size $B$.
  $m \leftarrow 0$
  **while** $m < M$ **do**
    Sample $\tilde{X}_1, \ldots, \tilde{X}_B \sim p_\theta(\,\cdot\,|X_m^p)$ {Batch sample}
    **for** $b = 1, \ldots, B$ **do**
      $\epsilon \sim \mathcal{N}(0, I)$ {Resample auxiliary variables}
      $X_b \leftarrow (X_m^p, \epsilon)$
      Sample $I_b \sim \text{Bernoulli}(\alpha(X_b, \tilde{X}_b))$
    **end for**
    **if** $S := \{b : I_b = 1, 1 \le b \le B\} \neq \emptyset$ **then**
      $a = \min(S)$ {First accepted sample}
      $(X_{m+1}^p, \ldots, X_{m+a-1}^p) \leftarrow (X_m^p, \ldots, X_m^p)$
      $X_{m+a}^p \leftarrow \tilde{X}_a^p$
      $m \leftarrow m + a$
    **else**
      $(X_{m+1}^p, \ldots, X_{m+B}^p) \leftarrow (X_m^p, \ldots, X_m^p)$
      $m \leftarrow m + B$
    **end if**
  **end while**
**output** $X_0^p, \ldots X_M^p$

---

**Algorithm 2** Fast, biased exploration of the state space with Timewarp

---

**Require:** Initial state $X_0^p$, number of steps $M$, maximum allowed energy increase $\Delta U_{\max}$
    **for** $m = 0, \ldots, M$ **do**
        Sample $\tilde{X}_m^p \sim p_\theta(\cdot \mid X_m^p)$ {Sample from conditional flow}
        **if** $U(\tilde{X}_m^p) - U(X_m^p) < \Delta U_{\max}$ **then**
            $X_{m+1}^p \leftarrow \tilde{X}_m^p$
        **else**
            $X_{m+1}^p \leftarrow X_m^p$ {Reject if energy change is too high}
        **end if**
    **end for**
**output** $X_0^p, \ldots X_M^p$

---

Note that unlike Algorithm 1, there is no need for the auxiliary variables, since the conditional flow only depends on the positions, and no MH acceptance ratio is computed here. The potential energy $U$ includes here also a large penalty if the ordering of a chirality center changes as described in Appendix A.3. As sampling proposals from Timewarp can be batched, we can generate $B$ chains in parallel, all starting from the same initial state. This batched sampling procedure leads to even further speedups. For all exploration experiments we use a batch size of 100, and run $M = 10000$ exploration steps. The maximum allowed energy change cutoff is set at $\Delta U_{\max} = 300 \text{kJ/mol}$.

## E  Dataset details

We evaluate our model on three different datasets, AD, 2AA, and 4AA, as introduced in Section 6. All datasets are simulated in implicit solvent using the `openMM` library [7]. For all MD simulations we use the parameters shown in Table 1.

Table 1: OpenMM MD simulation parameters

| | |
|---|---|
| Force Field | amber-14 |
| Time step | 0.5fs |
| Friction coefficient | $0.3\frac{1}{\text{ps}}$ |
| Temperature | 310K |
| Integrator | LangevinMiddleIntegrator |

We present more dataset details, like simulation times and number of peptides, in Table 2.

## F  Hyperparameters

Depending on the dataset, different Timewarp model sizes were used, as shown in Table 3. For all datasets the Multihead kernel self-attention layer consists of 6 heads with lengthscales $\ell_i = \{0.1, 0.2, 0.5, 0.7, 1.0, 1.2\}$, given in nanometers.

Table 2: Dataset details

| Dataset name | AD | 2AA | 4AA |
|---|---|---|---|
| Training set simulation time | 100 ns | 50 ns | 50 ns |
| Test set simulation time | 100 ns | 1 $\mu$s | 1 $\mu$s |
| MD integration step $\Delta t$ | 0.5 fs | 0.5 fs | 0.5 fs |
| Timewarp prediction time $\tau$ | $0.5 \times 10^6$fs | $0.5 \times 10^6$fs | $0.5 \times 10^5$fs |
| No. of training peptides | 1 | 200 | 1400 |
| No. of training pairs per peptide | $2 \times 10^5$ | $1 \times 10^4$ | $1 \times 10^4$ |
| No. of test peptides | 1 | 100 | 30 |

Table 3: Timewarp model hyperparameters

| Dataset | RealNVP layers | Transformer layers | Parameters | Atom-embedding dim $H$ |
|---------|----------------|--------------------|------------|-------------------------|
| AD | 12 | 6 | $1 \times 10^8$ | 64 |
| 2AA | 12 | 6 | $1 \times 10^8$ | 64 |
| 4AA | 16 | 16 | $4 \times 10^8$ | 128 |

The $\phi_{\text{in}}$ and $\phi_{\text{out}}$ MLPs use SiLUs as activation functions, while the Transformer MLPs use ReLUs. Note the transformer MLP refers to the atom-wise MLP shown in Figure 2, Middle inside the transformer block. The shapes of these MLPs vary for the different datasets as shown in Table 4.

Table 4: Timewarp MLP layer sizes

| Dataset | $\phi_{\text{in}}$ MLP | $\phi_{\text{out}}$ MLP | Transformer MLP |
|---------|------------|-------------|------------------|
| AD | $[70, 256, 128]$ | $[128, 256, 3]$ | $[128, 256, 128]$ |
| 2AA | $[70, 256, 128]$ | $[128, 256, 3]$ | $[128, 256, 128]$ |
| 4AA | $[134, 2048, 128]$ | $[128, 2048, 3]$ | $[128, 2048, 128]$ |

The first linear layers in the kernel self-attention module always has the shape $[128, 768]$ (in Section 4 denoted as $V$), and the second (after concatenating the output of head head) has the shape $[768, 128]$. The transformer feature dimension $D$ is for all datasets 128.

After likelihood training, we fine-tune the model for the AD and 2AA dataset with a combination of all three losses discussed in Section 5. We did not perform fine tuning for the model trained on the 4AA dataset. We use a weighted sum of the losses with weights detailed in Table 5. We use the

Table 5: Timewarp loss weighting factors

| Dataset | $\mathcal{L}_{\text{lik}}(\theta)$ | $\mathcal{L}_{\text{acc}}(\theta)$ | $\mathcal{L}_{\text{ent}}(\theta)$ |
|---------|------------------|------------------|------------------|
| AD | 0.99 | 0.01 | 0.1 |
| 2AA | 0.9 | 0.1 | 0.1 |
| 4AA | 1 | 0 | 0 |

*FusedLamb* optimizer and the *DeepSpeed* library [34] for all experiments. The batch size as well as the training times are reported in Table 6. All simulations are started with a learning rate of $5 \times 10^{-4}$,

Table 6: Timewarp training parameters

| Dataset + training method | Batch size | No. of A-100s | Training time |
|---------------------------|------------|---------------|---------------|
| AD — likelihood | 256 | 1 | 1 week |
| AD — acceptance | 64 | 1 | 2 days |
| 2AA — likelihood | 256 | 4 | 2 weeks |
| 2AA — acceptance | 256 | 4 | 4 days |
| 4AA — likelihood | 256 | 4 | 3 weeks |

the learning rate is then consecutively decreased by a factor of 2 upon hitting training loss plateaus.

# G   Computing infrastructure

The training was performed on 4 NVIDIA A-100 GPUs for the 2AA and 4AA datasets and on a single NVIDIA A-100 GPU for the AD dataset. Inference with the model as well as all MD simulations were conducted on single NVIDIA V-100 GPUs for AD and 2AA, and on single NVIDIA A-100 GPUs for 4AA.

