# OpenReview forum: "Timewarp: Transferable Acceleration of Molecular Dynamics by Learning Time-Coarsened Dynamics"
_NeurIPS.cc/2023/Conference — NeurIPS 2023 spotlight_

### Official Review · Reviewer_UaAm · 2023-06-28

**Soundness:** 3 good
**Presentation:** 4 excellent
**Contribution:** 2 fair
**Rating:** 7
**Confidence:** 4

**Summary:**

This paper addresses high-throughput MD simulations.
The proposed sampling algorithm simulates the MD of 10^5-10^6 fs timestep by a pre-trained model of a normalizing flow.
The normalizing flow can be used as a proposal distribution in an MCMC method.
The experimental results show the pre-trained model trained on peptides with two or four amino acids generalized to unseen combinations of amino acids.
The proposed method is faster than naive MD in many cases of two amino acids.

**Strengths:**

* The paper is well-organized and easy to follow.
* The proposed method can be integrated with a popular sampling method, MCMC.
* The batched sampling method compensates for the low acceptance ratio of the proposed model.
* The experimental results of the proposed exploration algorithm are also provided.
* No speed-up for no slow transitions is reasonable.
* The limitations are clearly shown in the experimental results.

**Weaknesses:**

* No comparison with conventional MD-acceleration methods such as metadynamics or parallel tempering. I would like to know how the proposed method is competitive with manually-tuned those methods.
* The proposed method needs to be evaluated on application-oriented metrics such as the reproducibility of some physical properties.
* Although the proposed method largely improved the simulation speed, the MH acceptance rates for the model are low, so the diffeomorphism function seems to need to be learned better.
* According to Figure 3, the proposed method cannot reproduce the transition states, i.e., the hill of MD's energy surfaces, so the method may not be applicable to chemical reactions.

**Questions:**

* Please provide the computational environment for the baseline MD simulation, amber-14. Is the wall clock comparison fair enough in terms of their implementation?
* Why is the proposed method slower on 4AA? Is this because of O(N^2) computational complexity of the attention mechanism or the size of the neural networks?

**Limitations:**

The limitations are clearly stated in Section 7.

---

> ### Author Rebuttal · Authors · 2023-08-09
>
> We thank the reviewer for their insightful review and questions. We now address their questions individually. To stay within the character limit for our rebuttals, we often cite parts of the questions.
>
> > No comparison with conventional MD-acceleration methods ...
>
> We do not compare to other MD-acceleration methods as Timewarp does not require any prior knowledge for unseen systems unlike common acceleration methods. Hence, we argue that the comparison provided in the paper is the fairest.
> However, Timewarp could also be accelerated by incorporating knowledge about the system into the algorithm. One possibility is to alternate Timewarp proposals with some (learned) change of collected variables, like the dihedral angles for peptides. These combined steps would still allow unbiased sampling from the target distribution.
> Timewarp could also be trained at multiple temperatures, by either training one model at each temperature or having the temperature as an additional conditioning input for a single model, which then could be applied similar to regular parallel tempering.
> We will include these aspects in the limitations section and the potential applications in the Conclusion.
>
> > The proposed method needs to be evaluated on application-oriented metrics such as the reproducibility of some physical properties.
>
> We utilize widely accepted evaluation metrics commonly found in the literature (Noé et al. Science 2019). These metrics not only adhere to established standards but also employ a level of rigor surpassing that of many other physical properties. Specifically, we evaluate energies alongside TICA plots, along with corresponding free energy projections. Additionally, we analyze bond lengths and autocorrelations, as detailed in appendix B.
>
> > ... the diffeomorphism function seems to need to be learned better.'
>
> The low acceptance rate is not a limitation in itself. There is a tradeoff between the achieved acceptance rate and the length of the time step. For smaller time steps the acceptance rate is usually higher and vise versa. Due to  Alg 1, we can evaluate multiple samples in parallel at nearly no additional costs. Hence, a lower acceptance rate in combination with a larger time step can be beneficial.
> In Figures 11 and 12 in appendix B.3, we show that the learned proposal distribution for unseen peptides matches the target closely. Moreover, the Timewarp exploration algorithm does not diverge. However, we agree that a more expressive proposal kernel would allow for larger time steps and ultimately result in even larger speed-ups compared to MD. We leave the exploration of more expressive architectures for future research.
>
> > the proposed method cannot reproduce the transition states, ..., so the method may not be applicable to chemical reactions.'
>
> We agree that this is a limitation of the proposed method and will include this in the limitations section. However, this is a common challenge faced by generative machine learning methods, as they often generate fewer samples compared to traditional MD simulations within the same timeframe. Consequently, this scarcity of samples becomes even more pronounced in transition states. Nevertheless, it is crucial to highlight that despite this limitation, our Timewarp MCMC method still allows us to infer unbiased free energy differences.
>
> > Please provide the computational environment for the baseline MD simulation...
>
> The used parameters for the MD simulations can be found in appendix E and G. We run the Timewarp Model as well as the MD simulations on the same type of GPUs for each experiment to allow for a fair wall-clock time comparison. For both we generate a single trajectory per GPU. A batched simulation is not available in existing fast MD libraries like OpenMM, as they are designed for simulating large systems serially. To parallelise, a custom implementation is needed to run separate simulations on each GPU thread. Also, if multiple trajectories were generated with MD, it is not trivial to combine these into a single unbiased trajectory and compute ESS/s, as we do now. If instead we were not concerned with correctly reweighting multiple trajectories, we could then run Timewarp in exploration mode (Alg 2), which easily produces 100 trajectories in parallel on a single GPU, which is comparable or greater than the number of parallel MD simulations which could be run in theory.
> We do not expect that our results would change drastically for using a different classical force field or MD library.
>
> > Why is the proposed method slower on 4AA? ...
>
> This is an excellent question. There is no one reason that the speed-up is slower for the 4AA system. The four main factors are the time step $\tau$, the acceptance rate, the forward pass through the flow, and the cost of the energy evaluation. As discussed earlier, there is a tradeoff between the time step and the acceptance rate. As described in appendix F the model for 4AA is about four times larger than for 2AA, resulting in an overall slower forward pass. As the attention weights are only computed once during each forward pass, the influence on the forward pass is marginal. The energy evaluation is slightly more expensive for 4AA compared to 2AA, which helps with the overall speed-up compared to MD as Timewarp requires about 100 times less energy evaluation per effective sample. Hence, by using a more expensive energy function, such as a semi-empirical force field, would probably improve the results significantly. We leave this to future work, as the MD simulations for evaluation purposes become orders of magnitude more expensive.
> In case of the 4AA system the most important factor for the lower speed-up is the smaller time step, which was required to achieve acceptance rates around 1%, as shown in appendix B. More tuning could result in better performance.
> We will add parts of this discussion to the limitations and conclusion section.

---

> > ### Comment · Reviewer_UaAm · 2023-08-19
> >
> > I appreciate the response from the authors. The answers to my questions were satisfying.
> >
> > While there are still many challenges to practical applications, the contribution of this paper is substantial. I believe this paper should be accepted, and I have raised my score.

---

### Official Review · Reviewer_fomi · 2023-07-01

**Soundness:** 3 good
**Presentation:** 3 good
**Contribution:** 3 good
**Rating:** 7
**Confidence:** 3

**Summary:**

In this paper the authors introduce Timewarp, a method that relies on a normalizing flow as a proposal distribution for enhanced sampling of the Boltzmann distribution. The key results include an acceleration in the MD time step from ~1 fs to 10^5 fs and transferability between peptide systems.

**Strengths:**

The core demonstration of generalization to unseen peptides in an all-atom representation is an important contribution.

The method is clearly presented and the value of accelerated sampling is also clear.


**Weaknesses:**

The method seems to break down already for 4-mer peptide systems. The limitations in expressivity of normalizing flows shown throughout the literature cast doubt on the scalability of the method to practical applications.

Performance benchmarks with respect to classical MD approaches are difficult to evaluate, due to choices in force field, integrator, etc.

**Questions:**

Can the authors expand on the choice to treat velocity as a non-physical auxiliary variable? It is clear that it makes a permutation equivariant coupling layer easier to construct, but is there no information loss that might contribute to the low MH acceptance rate?
How many heads and how were length scales determined for each head in the kernel self-attention? Does ablating this tell you anything about the importance of including long-range interactions, or is the locality assumption implicit in the kernel self-attention sufficient for capturing the dynamics for the peptides considered?
Can the authors provide additional references and discussion (perhaps in the appendix) of the evaluation metric presented in Eq 18? What ESS/s values are reported when comparing different classical MD algorithms?
Since Timewarp is evaluated on small peptide systems, conformer generation with a semi-empirical method like xTB might also be an appropriate evaluation. Can the authors discuss their method in the context of such approaches?
Are there any concrete proposals for future work that would address the current limitations in MH acceptance rate, system size, and the troubling trends in scaling to larger systems?
Why do the authors use amber-14 in openMM rather than a more modern force field?
The caption in Fig 4 does not match the figure labels.
In Fig 3c the red dot indicating initial state is difficult to see.


**Limitations:**

Yes, although further discussion of possible avenues for addressing these limitations would be welcome.

---

> ### Author Rebuttal · Authors · 2023-08-09
>
> We thank the reviewer for their insightful review, questions, and suggestions. We appreciate their judgement that our paper is an important contribution. We now address their questions individually.
> To keep the rebuttal within the character limit, we often only cite parts of each question.
>
> > The method seems to break down already for 4-mer peptide systems...
>
> We agree on the limitation. However, generative flow models for molecular systems remain an active research field. New architectures might increase the expressiveness allowing scaling to larger systems.
> In our study, we primarily focused on addressing the transferability challenge, rather than scalability, as prior works in generative modeling for molecules in Cartesian coordinates had not demonstrated transferability before our research.
> Moreover, we  present also the Timewarp exploration algorithm, which does not include the Metropolis acceptance step, and shows significant speed-up over classical MD at the cost of not being asymptotically unbiased. Another common way to scale generative models to larger systems is coarse graining, which could be also used to scale Timewarp to larger systems.
>
> > Performance benchmarks with respect to classical MD approaches are difficult to evaluate, due to choices in force field, integrator, etc.
>
> We conducted our MD simulations using the widely-used openMM library, employing the popular Amber-14 force field, which is a common choice for unconstrained MD simulations. The simulation parameters we utilized can be found in appendix E and are standard for such simulations.
> We are confident that our results would not be significantly impacted if alternative MD libraries or classical force fields were employed. However, semi-empirical or other computationally expensive force fields, might yield better results compared to MD. See below for a more detailed discussion.
>
> > Can the authors expand on the choice to treat velocity as a non-physical auxiliary variable? ...
>
> We initial tried using the physical velocities, but found that replacing them with
> independent auxiliary variables that target a standard Normal worked better. This is allowed because we are only interested in simulating the position distribution, and the density in Eqn 4 has the same marginal for the positions as the non-augmented density in Eqn 1. The augmented velocities are still required to compute the acceptance ratio, though. Hence, there is no information loss and it should not affect the acceptance rate.
>
> > How many heads and how were length scales determined for each head in the kernel self-attention? ...
>
> We performed a manual hyperparameter search to determine the appropriate length scales for our Gaussian kernels. The finalized length scales used in our model can be found in appendix F. The usage of Gaussian kernels allows even particles that are far apart to contribute, enhancing its capability to capture relevant information from various distances. We discovered that including both short length scales (approximately 0.1nm) and long length scales (around 1nm) proved to be beneficial for the overall performance of our model.
>
> > Can the authors provide additional references and discussion (perhaps in the appendix) of the evaluation metric presented in Eq 18? ...
>
> The effective sample size (ESS) is commonly used in the literature to evaluate Markov chains.  A good summary of the ESS is given in the widely used STAN reference manual in 16.4.1. As we are interested in a wall-clock time comparison of our Timewarp algorithms and MD, we compare the ESS per second wall-clock time. To evaluate this metric, we need the sampling wall-clock time as well as the autocorrelations for different lag times (see Eqn. 18). Some example autocorrelation plots are reported in appendix B. We will extend the discussion in the final version of the manuscript.
>
> > Since Timewarp is evaluated on small peptide systems, conformer generation with a semi-empirical method like xTB might also be an appropriate evaluation...
>
> Using the semi-empirical xtb force field instead of a classical one, would probably yield larger speed-ups compared to MD, as the evaluation of the semi-empirical force field is much more expensive and the Timewarp sampling algorithm requires significantly fewer (about 100 times) energy evaluations. Moreover, some of these energy evaluations can be performed in parallel using the batch sampling algorithm (Alg. 1). Hence, the more expensive the energy function, the more attractive the Timewarp method becomes. We thank the reviewer for bringing this up and will include this idea for future research directions in the Conclusion section. One reason we did not try this is that the data generation and especially the evaluation become orders of magnitude more expensive.
>
> > Are there any concrete proposals for future work that would address the current limitations ...'
>
> The low acceptance rate is not a limitation in itself if the changes proposed are large enough that the effective sample size per second is fast for the model. The acceptance rate can be easily increased by predicting a smaller amount of time in the future, (e.g. if we set $\tau$ near zero, we can get high acceptance rates just by predicting $x(t+ \tau) = x(t)$) but this most likely results in a slower speed-up.
> Scaling to larger systems is often done by considering a coarse grained representation.
> We will include this in the limitations section.
>
> > Why do the authors use amber-14 in openMM rather than a more modern force field?
>
> We use the amber-14 force field as it is commonly used in the literature and implemented in openMM. We do not think that other classical force fields would change our findings significantly.
>
> > The caption in Fig 4 does not match the figure labels. In Fig 3c the red dot indicating initial state is difficult to see.
>
> Thanks for pointing this out, we will incorporate the suggested changes in the final version.

---

> > ### Comment · Reviewer_fomi · 2023-08-11
> >
> > I have read the rebuttal and am satisfied with the response.

---

### Official Review · Reviewer_heyF · 2023-07-07

**Soundness:** 3 good
**Presentation:** 3 good
**Contribution:** 4 excellent
**Rating:** 8
**Confidence:** 4

**Summary:**



This paper presents a Boltzmann generator for small molecular systems in implicit solvent.  The generator works demonstrates transferability to test systems of the same class (dipeptides or tetrapeptides).  It is the first Boltzmann generator that operates in Cartesian coordinates and demonstrates transferability.  The authors estimate that for these very small molecular systems in their study, the sampling via a Boltzmann generator could provide wall-time acceleration compared to vanilla MD simulation.


**Strengths:**

This paper is the first demonstration of a Boltzmann generator in Cartesian coordinates that generalizes to untrained systems.  This is certainly commendable, despite the existence of previous generators for small molecules that relied on internal coordinates.  The method is presented in a clear fashion and the authors plan to release their codes and data upon acceptance.  The experiments are sensible and show that the slow modes of the MD simulations are covered with a similar distribution by the sampler.  This paper is adding to a series of developments in the line of Boltzmann generators, an area that shows great promise, even though it is still at its infancy with respect to practical applications in drug discovery. As such, I think that it is a significant contribution to this conference.


**Weaknesses:**


The correct operation of this generator requires a Metropolis Hastings step, which will limit the application of the current technology to only very small systems.  This is not necessarily a problem with all Boltzmann generators, but this limitation is very clearly demonstrated in the work and in leads to a reduction in the acceptance rate as the number of particles doubled.  The authors don't discuss this possible limitation of the current work in their paper, nor suggest ways to think about scaling to systems of practical interest.

The work is only dealing with molecules in implicit solvent.  These implicit solvent simulations are not of practical use in drug discovery, so it would be useful to have a discussion of possible generalizations towards explicit simulations (either generating the full environment, or at least learning from such simulations and generating whatever is feasible).

It is hard to judge reproducibility without the availability of the code/data; I hope this will be alleviated once the paper is published.


**Questions:**


The work is limited to molecules in implicit solvent.  How difficult would it be scale these methods to also generate the full environment of water, ions, lipids and small molecules in typical molecular dynamics simulations in the context of drug discovery?  Perhaps in a more practical fashion, are there any limitations in the current architecture

Did the authors try to train the model on explicit solvent simulations and use only the non-water/non-ion coordinates in their systems?  The generation of the training trajectories does not seem particularly time consuming compared to the rest of the compute effort that went into the project and the possible wall-time improvements might be even larger (though of course this depends on the software and hardware that is employed).  The very limited number of total atoms in the current models would suggest that existing enhanced sampling methods for MD simulations (including those that include kinetic considerations) would also work for these small systems; however, such sampling techniques might be trickier to optimize in explicit water than in implicit water.  (Even without any sophisticated techniques, I would guess that the implicit-solvent dipeptides and tetrapeptides would reduce their autocorrelation much more quickly by sampling them at ridiculously high temperatures (or fluctuating temperatures as in simulated tempering), and could then be projected back to the baseline with an MH criterion.)

Why did the authors not mention the Torsional Diffusion paper in their introduction?  Wasn't that work also a generalizable Boltzmann generator (using internal coordinates), and probably the first of this kind?


**Limitations:**

No potential negative societal impact

---

> ### Author Rebuttal · Authors · 2023-08-09
>
> We thank the reviewer for their insightful review and questions. We appreciate their judgement that our paper is a significant contribution to this conference. We now address their questions individually.
>
> > The correct operation of this generator requires a Metropolis Hastings step, which will limit the application of the current technology to only very small systems... The authors don't discuss this possible limitation of the current work in their paper, nor suggest ways to think about scaling to systems of practical interest.
>
> We agree that the low speed-up for larger systems is a limitation of our work. We note that our limitations section clarifies that we only succeed in speeding up sampling for a third of the 4AA peptides.
> We leave the problem of scaling to larger systems to future work. A number of routes are possible. We list a few here:
> - We could remove the Metropolis-Hastings correction and focus solely on the exploration algorithm. This would allow us to scale beyond 4AA at the expense of potentially biased simulations. However, we believe this could still be extremely useful as long as methods were developed to validate the correctness of the predictions.
> - Removing the MH correction would also allow for coarse-graining, e.g., representing only all the heavy atoms, or only all the backbone atoms, instead of every single atom in the system.
> - Parameterisations such as those used in AlphaFold2 could also improve performance. There the backbone heavy atoms are represented as rigid frames. Hence the model would not need to learn these kinematic relationships, but these would be hard-coded into the representation.
>
> We will include a more detailed discussion of this in the final version.
>
> > The work is only dealing with molecules in implicit solvent...
>
> This is indeed a less realistic scenario than explicit solvent simulations. We focused on implicit solvent simulations in this paper in order to begin with the simplest scenario, and leave the question of explicit solvent to future work. In order to adapt Timewarp for explicit solvent simulation, the most straightforward extension would be to simply add the solvent atoms as additional nodes to be processed by the transformer. This would increase the computational cost of the forward pass. To mitigate this, we could investigate sparse transformer architectures/graph neural networks.
>
> > It is hard to judge reproducibility without the availability of the code/data
>
> We will release code and data publicly upon publication.
>
> > Why did the authors not mention the Torsional Diffusion paper in their introduction?
>
> We will include the Torsional Diffusion paper in the related work section. As mentioned by the reviewer, the main differences to our work is that they rely on internal coordinates and do not operate in the all atom system.

---

> > ### Comment · Reviewer_heyF · 2023-08-15
> > **Thank you for the response**
> >
> > I acknowledge having read the response by the authors and I remain positive about this work.
> >
> > I agree with the view that excluding the Metropolis Hastings step might be a feasible way forward.  In addition, I wonder if perhaps there is more to learn/mimic from existing methods for enhanced sampling.
> >
> > Finally, I suspect that the work on the inclusion of explicit solvent, although technically feasible, is qualitatively different from the present work, as it would have to deal with changes in the loss functions to account for permutations and proper hydrogen bonding.  I will look forward to the future work on this subject.

---

> > > ### Author Response · Authors · 2023-08-21
> > >
> > > We appreciate the reviewer's continued positivity and support for our work.
> > >
> > > > I agree with the view that excluding the Metropolis Hastings step might be a feasible way forward. In addition, I wonder if perhaps there is more to learn/mimic from existing methods for enhanced sampling.
> > >
> > > One possibility is to alternate Timewarp proposals with some (learned) change of collected variables, like the dihedral angles of peptides. These combined steps would still allow unbiased sampling from the target distribution. Timewarp could also be trained at multiple temperatures, by either training one model at each temperature or having the temperature as an additional conditioning input for a single model, which then could be applied similar to regular parallel tempering.
> > >
> > > > Finally, I suspect that the work on the inclusion of explicit solvent, although technically feasible, is qualitatively different from the present work, as it would have to deal with changes in the loss functions to account for permutations and proper hydrogen bonding. I will look forward to the future work on this subject.
> > >  research.
> > >
> > > We agree with the reviewer's insight regarding the potential challenges posed by explicit solvent systems. However, our Timewarp model is already well-suited for addressing these challenges. In the case of likelihood-based training, adapting solely involves modifying the training data by using trajectories of explicit solvent systems, thanks to the model's inherent permutation equivariance. For energy-based training and the Metropolis-Hastings step, we require force fields for explicit solvent models, such options are readily available. In alignment with the reviewer's assessment, we share the belief that this represents a promising avenue for future research.

---

### Official Review · Reviewer_Pymp · 2023-07-07

**Soundness:** 3 good
**Presentation:** 4 excellent
**Contribution:** 4 excellent
**Rating:** 8
**Confidence:** 5

**Summary:**

In this work, the authors present a method for speeding up MD simulations. The speeding up of MD simulations is needed to be able to simulate processes that occur over longer timespans (eg. protein folding). The method proposed by the authors, Timewarp, does so by learning the transition probability of MD simulations over timesteps beyond those normally used. The transition probability is learned using a normalising flow based on a Transformer architecture. Transition steps sampled using the normalising flow are pushed through a Metropolis-Hasting acceptance step to assure asymptotic unbiased samples. Additionally, the authors propose to run the normalising flow without the MH step for faster sampling. The normalising flow is learned using trajectories of multiple structures using traditional MD simulation. By using multiple structures at the same time, the authors aim to make the method generalize to unseen structures. This is one of their major contributions.

The work experimentally shows the success of the method on 3 different sizes of molecules; 1) Alanine Dipeptide, 2) 2 amino acid peptides and 3) 4 amino acid peptides. The experiments show that

-Timewarp closely matches the samples from MD
-Timewarp exploration (without MH step) provides estimates of the free energy close to MD, but less accurate then MCMC (with MH step).



**Strengths:**

NOTE: This is an updated review from a review round of an earlier conference. The review is updated to reflect the changes made by.

**Originality:** The paper discusses a very interesting new intermediate problem in ML for Chem/Bio that lies in between the two more often studied problems of Force Field learning and Conformer generation. I've commented on this below.

**Quality:** The paper provides a good introduction to the problem at hand and as such sufficiently motivates the work. The proposed solution is interesting and makes good use of existing methods for similar problems without overcomplicating it. Experiments are in depth and well executed.

**Clarity:** The paper is well written and structured. Figures are of high quality.

**Significance:** The paper has the potential to be very significant. The problem introduced is novel and important.

**Weaknesses:**

- Unfortunately, there's no code available and the datasets are not open sourced.
- Related work could be improved with more references to work from the computational chemistry literature on enhanced sampling(ie. topic such as Transition Path Sampling and Bias Potential methods).


**Questions:**

As I commented above, from my understanding, the problem discussed and addressed in this paper lies somewhere in between force field learning and conformer generation. The transition step length tau in this case determines which of the two the problem is closer to. If tau is very small, the problem at hand is similar to force field learning. With tau very large, the conditional distribution becomes similar to learning the Boltzmann distribution. Considering this, I'm surprised the work does not contain a study of the influence of tau on the speedup seen by the method over MD. Could the authors comment on this? For example, why was the specific value for tau used in the paper chosen?

**Limitations:**

The authors have sufficiently addressed the limitations.

---

> ### Author Rebuttal · Authors · 2023-08-09
>
> We thank the reviewer for their insightful review, and appreciate their assessment that the paper has the potential to be very significant. We now address their comments individually.
>
> > Unfortunately, there's no code available and the datasets are not open sourced.
>
> We will release the code and datasets publicly upon publication of the paper. Unfortunately we cannot release our code and datasets anonymously before publication.
>
> > Related work could be improved with more references to work from the computational chemistry literature on enhanced sampling'
>
> Thank you for pointing us to the topics of transition path sampling and bias potential methods. We will include a discussion of this in the related work section in the final version. We believe these could both in practice be used in conjunction with a method like Timewarp in order to further accelerate sampling. One possibility is to alternate Timewarp proposals with some (learned) change of collected variables, like the dihedral angles of peptides. These combined steps would still allow unbiased sampling from the target distribution. Timewarp could also be trained at multiple temperatures, by either training one model at each temperature or having the temperature as an additional conditioning input for a single model, which then could be applied similar to regular parallel tempering. We will include these possible directions for future research in the Conclusion.
>
> > I'm surprised the work does not contain a study of the influence of tau on the speedup seen by the method over MD. Could the authors comment on this? For example, why was the specific value for tau used in the paper chosen?'
>
> The impact of tau on the speedup seen by the method over MD is indeed an important research question to address. We found that the choice of tau involved a trade-off: if tau was too large, then the prediction problem becomes more difficult. This can lead to low acceptance rates, and hence a smaller speedup. However, if tau is too small, then the model must be run many times in order for the Markov chain to converge, again leading to a smaller speedup. As we introduced the batch acceptance algorithm (Algorithm 1), we can evaluate multiple samples in parallel at nearly no additional costs. Hence, a lower acceptance rate in combination with a larger time step is usually preferable, and an acceptance rate of nearly one would waste the potential for larger time steps. The current values were chosen by manual hyperparameter search guided by these intuitions. Unfortunately, a systematic hyperparameter search by trying a grid of tau values was beyond the computational resources of our project. We will add a further discussion of this to the final version of the paper.

---

> > ### Comment · Reviewer_Pymp · 2023-08-16
> > **Score increase**
> >
> > Thank you for your response. With the proposed addition of a discussion on enhanced sampling and a further discussion on the role of tau in mind, I will increase my score.

---

### Author Rebuttal · Authors · 2023-08-07

We thank all the reviewers for taking the time to review our paper.
A number of authors have asked for code and datasets to be released. We will release the code upon publication under an open source license. We will also publicly release the datasets that we used to train and evaluate the models. Unfortunately we cannot release our code and datasets anonymously before publication.

---

### Decision · Program_Chairs · 2023-09-21

**Decision:**

Accept (spotlight)

**Comment:**

All reviewers really liked this paper so this is clearly an accept.

One may also perhaps argue for a spotlight presentation. However, this AC thinks that the method has the limitation that it does not allow for estimation of physical observables such as folding rates. So there are still room for improvement.